# Cancer Metastasis and Treatment Resistance: Mechanistic Insights and Therapeutic Targeting of Cancer Stem Cells and the Tumor Microenvironment

**DOI:** 10.3390/biomedicines10112988

**Published:** 2022-11-21

**Authors:** Ethan J. Kilmister, Sabrina P. Koh, Freya R. Weth, Clint Gray, Swee T. Tan

**Affiliations:** 1Gillies McIndoe Research Institute, Wellington 6242, New Zealand; 2Wellington Regional Plastic, Maxillofacial & Burns Unit, Hutt Hospital, Lower Hutt 5010, New Zealand; 3Department of Surgery, The Royal Melbourne Hospital, The University of Melbourne, Parkville, VIC 3010, Australia

**Keywords:** cancer stem cell, metastasis, treatment resistance, tumor microenvironment, renin-angiotensin system

## Abstract

Cancer metastasis and treatment resistance are the main causes of treatment failure and cancer-related deaths. Their underlying mechanisms remain to be fully elucidated and have been attributed to the presence of cancer stem cells (CSCs)—a small population of highly tumorigenic cancer cells with pluripotency and self-renewal properties, at the apex of a cellular hierarchy. CSCs drive metastasis and treatment resistance and are sustained by a dynamic tumor microenvironment (TME). Numerous pathways mediate communication between CSCs and/or the surrounding TME. These include a paracrine renin-angiotensin system and its convergent signaling pathways, the immune system, and other signaling pathways including the Notch, Wnt/β-catenin, and Sonic Hedgehog pathways. Appreciation of the mechanisms underlying metastasis and treatment resistance, and the pathways that regulate CSCs and the TME, is essential for developing a durable treatment for cancer. Pre-clinical and clinical studies exploring single-point modulation of the pathways regulating CSCs and the surrounding TME, have yielded partial and sometimes negative results. This may be explained by the presence of uninhibited alternative signaling pathways. An effective treatment of cancer may require a multi-target strategy with multi-step inhibition of signaling pathways that regulate CSCs and the TME, *in lieu* of the long-standing pursuit of a ‘silver-bullet’ single-target approach.

## 1. Introduction

Metastasis, a hallmark of cancer, is a phenomenon in which cells from a primary tumor form new tumors in regional lymph nodes and/or at distant sites [1]. It accounts for up to 90% of cancer-related deaths [2]. However, the mechanisms underlying cancer metastasis remain poorly understood [1,3,4,5]. Despite the significantly growing list of potential therapeutic targets, cancer remains a largely unsolved problem. Current treatment strategies for cancer include surgery, chemotherapy, radiotherapy, and immunotherapy, often in combination, which are associated with varying rates of treatment failure, manifesting as loco-regional recurrence, and/or distant metastasis. Improved understanding of the mechanisms driving metastasis and treatment resistance, will enable advances in developing a durable treatment of cancer.

The stochastic model of cancer, also known as the clonal evolution theory of cancer, largely encapsulates features of Darwinian evolution. In this model, cancer cells undergo stepwise genetic and epigenetic changes, with those acquiring advantageous mutations undergoing clonal selection and expansion within a specific tumor microenvironmental *niche* [6,7] (Figure 1A). Conventional cancer therapies, which create a selection pressure for the clonally expanded cancer cell population, may inadvertently drive clonal expansion of therapy-resistant clones [8].

The hierarchical model of cancer, also known as the cancer stem cell (CSC) concept, proposes that tumorigenesis is driven by CSCs—a small subpopulation of quiescent cancer cells imbued with pluripotency and self-renewal properties [10] (Figure 1B). CSCs undergo symmetric cell division to give rise to identical CSCs, and asymmetric division to give rise to differentiated cancer cells that comprise the bulk of the heterogenous tumor cell population [10]. Failure of conventional treatments of cancer—surgery, radiotherapy, chemotherapy, and immunotherapy, may result from CSC traits that enable evasion of these treatments. Proposed mechanisms for treatment resistance include: tumor dormancy, presence of drug efflux pumps [11,12,13,14], up-regulation of pro-survival pathways [15] and anti-apoptotic proteins [12,14], enzymes counteracting oxidative stress [13], expression of DNA repair mechanisms [13,14,16], and dysregulated expression of long non-coding RNAs (lncRNA) [17,18]. These proposed mechanisms may enable CSCs to escape cell death from conventional treatments, accounting for loco-regional recurrence and distant metastasis. Appreciation of these mechanisms will lead to improved therapeutic strategies to target this CSC population [10,19]. An emerging concept relating to treatment resistance is multi-drug resistance (MDR). This concept describes resistance to multiple structurally unrelated therapies, such as chemotherapy, radiotherapy, immunotherapy, and hypoxia, and occurs in up to 70% of malignancies at the time of diagnosis [20]. Prevalence of MDR increases after initial treatment with chemotherapy due to selection pressures for drug resistant traits in tumor cells [20]. Thus, the concept of MDR essentially summarizes a chemo-immune-radiotherapy resistant phenotype, and appreciation of all avenues leading to this multi-resistant phenotype will help facilitate a multi-faceted approach to the effective treatment of cancer.

Cancer metastasis involves four steps, known collectively as the metastatic cascade. These steps include (1) detachment of cancer cells from the primary tumor and invasion through the basal lamina, (2) intravasation of the cancer cells into the circulatory system, (3) survival of cancer cells within the vasculature, and (4) extravasation, colonization and growth within the distant metastatic site [21,22]. The phenotype and abilities of CSCs, particularly their plasticity and subsequent ability to adapt to different tissue microenvironments, sets a favorable foundation for metastasis. The ability of CSCs to transition between mesenchymal-like and epithelial-like phenotypes enables them to adapt to the broad requirements of each stage of metastasis [23].

CSCs are sustained by a complex and dynamic tumor microenvironment (TME) comprised of cellular and non-cellular components. Cellular components of the TME include CSCs and their downstream progenies, immune cells, mesenchymal stem cells, stromal cells, pericytes, adipocytes and cancer-associated fibroblasts. Non-cellular elements consist of extra-cellular matrix (ECM) components. Numerous systems and signaling pathways contribute to facilitating intercellular communication and maintenance of a tumorigenic *niche*. These include the immune system, the paracrine renin-angiotensin system (RAS), and the Wnt/β-catenin, Notch, and Sonic Hedgehog (SHH) signaling pathways.

This review presents mechanistic insights into cancer metastasis by exploring the role of CSCs and key components of the TME in contributing to tumor growth, metastasis, and treatment resistance. We explore potential therapeutic targets in the treatment of cancer, including a novel multimodal strategy by modulation of CSCs, the surrounding TME, and critical signaling pathways.

## 2. Cancer Stem Cells in Tumor Growth, Metastasis, and Treatment Resistance

### 2.1. Cancer Stem Cells

According to the stochastic model of cancer (Figure 1A) cells acquiring advantageous mutations undergo clonal expansion to form a homogenous tumorigenic population [6,7]. Based on principles of Darwinian evolution, mainstay treatments such as radiotherapy, chemotherapy and immunotherapy exert selection pressures and expansion of treatment resistant clones. This perpetuates treatment resistance and treatment failure.

The hierarchical model of cancer (Figure 1B) proposes that tumorigenesis is sustained by CSCs—a small subpopulation of tumorigenic cancer cells, imbued with features of pluripotency, self-renewal, and the ability to evade tissue homeostatic mechanisms including apoptosis, resulting in aberrant growth [6,24,25,26,27,28,29]. These CSCs undergo symmetric division giving rise to CSCs, and asymmetric division which generates progenitor cells capable of undergoing terminal differentiation to form non-tumorigenic cancer cells [24,25,26,30,31].

First demonstrated in acute myeloid leukemia (AML) [32,33], CSCs have been identified in many types of solid cancers, such as cutaneous squamous cell carcinoma (SCC) [34], head and neck SCC [35,36] including oral cavity SCC affecting different subsites [37,38,39], metastatic malignant melanoma [40,41], glioblastoma [42,43] and other brain tumors [44,45], renal clear cell carcinoma [46], breast [47], colorectal [48], gastric [49], liver [50], lung [51], and prostate [52] cancers.

The precise origin of CSCs has yet to be fully elucidated, with current theories proposing a possible origin from (1) the accumulation of genetic and epigenetic changes amongst resident tissue stem cells [27]; (2) reprogramming of adult stem cells back to an induced pluripotent state [27,53]; (3) and through epithelial-mesenchymal transition (EMT) by which differentiated cancer cells de-differentiate back to a pluripotent state [27]. The CSC hierarchy is a dynamic multi-directional system, with non-CSCs capable of de-differentiating back into a pluripotent state. This phenomenon is known as plasticity, which adds further challenges to effective treatment of cancer [54,55,56,57]. The presence of a dynamic plastic CSC hierarchy has been demonstrated in many types of human cancers including glioblastoma [54], breast [55,56,57] and colorectal [58] cancer.

The stochastic model of cancer and the hierarchical model of cancer are not mutually exclusive, and tumorigenesis is likely to embody concepts from both models, with the CSCs able to undergo mutations to adapt to the TME, as featured in the stochastic model. The concept of plasticity, further complexifies cancer biology and may add further intricacies to the concept of treatment resistance. CSCs, the small self-renewing population within cancers, hitherto remains an elusive therapeutic target in the treatment of cancer. Effective targeting of this population requires appreciation of the various mechanisms by which this self-sufficient population evades current treatments.

### 2.2. Cancer Stem Cells and Metastasis and Treatment Resistance

Tumors are comprised of a phenotypically diverse hierarchical population of cells. While CSCs constitute a small proportion of the tumor cell population, they are a compelling therapeutic target due to their ability to sustain tumor growth through their pluripotency and self-renewal capacities that contribute to tumor recurrence and metastasis. Proposed mechanisms by which CSCs may contribute to treatment resistance include tumor dormancy, expression of drug efflux pumps, activation of DNA repair mechanisms, dysregulated expression of lncRNAs, and up-regulation of pro-survival pathways, dehydrogenase (ALDH) activity and expression of anti-apoptotic proteins such as B-cell lymphoma-2 (BCL2).

Tumor dormancy describes a phenomenon by which CSCs enter a dormant state and reside in the G_0_-phase of the cell cycle, thus evading the actions of anti-cancer drugs and radiotherapy, which primarily target rapidly dividing cells [14]. Under the right stimuli, these dormant CSCs can be recruited to re-enter the cell cycle, thus giving rise to loco-regional recurrence and distant metastasis. Other mechanisms by which CSCs can engage in tumor dormancy include EMT and its reverse process, mesenchymal-to-epithelial transition (MET) [14].

Expression of drug efflux pumps, also known as ATP-binding cassette (ABC) transporters, may contribute to chemotherapy resistance [11,12,13,14]. Enhanced expression of these transporters has been observed in various cancer types including breast, bladder, lung [59] and ovarian cancer, and AML [60], and has been shown to be expressed by CSCs [59,61,62]. These pumps allow rapid efflux of cytotoxic agents, thus contributing to chemotherapy resistance.

Chemotherapy and radiotherapy induce cell death through various mechanisms of DNA damage. In normal cells, stringent DNA repair mechanisms exist, to prevent formation of carcinogenic mutations. If these mutations cannot be amended, these cells are programmed to undergo apoptosis. In contrast, CSCs, have up-regulated expression of DNA repair mechanisms, allowing evasion of lethal DNA damage [11,13,14]. One of these mechanisms is through activation of a family of checkpoint kinases (Chk) including ATM-ChK1 and ATR-ChK2. These kinases act as DNA checkpoints which halt cell cycle progression, to facilitate DNA repair [13]. Interestingly, this DNA checkpoint repair system has been shown to be more efficient in stem-cell like populations in lung cancer and glioblastoma [63,64], and a mouse model of breast cancer [65].

Both chemotherapy and radiotherapy lead to the generation of reactive oxygen byproducts which can induce cellular damage. ALDH are a family of enzymes involved in cellular responses to oxidative stress through production of NADPH, an antioxidant, which acts as a free radical scavenger system [13]. Increased expression of ALDH may protect CSCs against the harmful byproducts of radiotherapy and chemotherapy [13].

Survival of CSCs is further promoted by evasion of apoptosis. This is achieved through up-regulation of anti-apoptotic pathways and proteins, including the BCL-2 family [12,14]. BCL-2 has been observed to be up-regulated in CSCs in HCC [66]. Conversely, its down-regulation has been associated with improved chemosensitivity of CSCs in colorectal cancer [67]. Various other pro-survival pathways are implicated in CSC maintenance and treatment resistance including the Notch pathway [12,68] via its role in cell growth and metastasis. Enhanced Notch signaling has been observed in stem cell-like population in lung cancer [69], glioma [70,71], and ovarian cancer [72]. Other signaling pathways implicated in promoting CSC growth and treatment resistance include the Wnt/β-catenin and SHH pathway [73].

Expression of lncRNA, areas of non-coding RNA involved in the regulation of gene expression, may further contribute to chemotherapy and immunotherapy resistance. This is postulated to occur though the role of lncRNA in maintaining CSCs, promoting EMT, inhibiting apoptosis, and regulating the surrounding TME by exporting lncRNA through exosomes [17,18].

Ultimately, these mechanisms confer therapy resistance, confound effective treatment of cancer, and the inability to abolish the CSC population leads to loco-regional recurrence, and distant metastasis.

### 2.3. Cancer Stem Cells and Multi-Drug Resistance

There are multiple mechanisms by which CSCs evade conventional treatments. The concept of treatment resistance is further complexified by the emerging phenomenon of MDR, which highlights a phenotype by which CSCs simultaneously exhibit resistance to multiple unrelated drugs, in particular various chemotherapeutic agents [20]. These MDR cells are often simultaneously resistant to multiple other treatment modalities including radiotherapy, immunotherapy, and hypoxia [20]. The most common mechanism by which MDR occurs is through the expression of the aforementioned drug efflux pumps, also known as ABC transporters [74]. The ABC family of proteins is encoded by 48 genes and facilitates translocation of various agents including chemotherapeutic drugs. P-glycoprotein (Pgp) is one of the key members of the ABC-transporter family implicated in MDR. In MDR, the most relevant Pgp transporters are multidrug resistance associated-protein 1 (MRP1, also known as ABCC1) and ABCG2 [74]. Heat shock proteins (HSPs), a molecular chaperone, may contribute to MDR. Chaperones are implicated in many cellular events, and are involved in epigenetic regulation of gene expression [75]. HSP90 is a specific chaperone implicated in the differentiation, invasion and metastasis of cancer cells, and may be involved in MDR through its effect on Pgp expression [75]. Other mechanisms which are employed by CSCs which contribute to MDR include loss of pro-apoptotic factors, enhanced DNA repair mechanisms, the ability to metabolize anti-cancer drugs to less active or inactive metabolites and sequestration of drugs in organelles separate from their target [75]. Ability to modulate the various mechanisms contributing to MDR will enhance chemo-sensitivity of CSCs.

## 3. Mechanisms of Metastasis

### 3.1. The Metastatic Cascade

Cancer metastasis is proposed to occur by sequential steps, known as the metastatic cascade, whereby cancer cells disseminate to loco-regional or distant sites to recapitulate the original tumor. These steps include: (1) detachment of cancer cells from the primary tumor and invasion through the basal lamina, (2) intravasation of cancer cells into the circulatory system, (3) survival of cancer cells within the vasculature as circulating tumour cells (CTCs), and (4) when CTCs undergo MET that enables extravasation, colonization and growth within the distant metastatic site [21,22] (Figure 2).

During metastasis most disseminated cancer cells perish, leaving a minute proportion of cancer cells that possess metastatic ability [76]. Not all disseminated tumor cells possess metastatic potential, as they may be too differentiated or senescent. Disseminated CSCs are pluripotent and are capable of asymmetric cell division, indefinite self-renewal, and demonstrate plasticity in response to the surrounding changing tissue microenvironmental *niche*. Upon arrival at the distant site, these disseminated cancer cells may enter a period of latency, then may undergo reactivation and formation of a macroscopic metastatic tumor [77], resembling the parent tumor. These observations support the hypothesis that CSCs drive metastasis.

### 3.2. Epithelial-to-Mesenchymal Transition

One of the key pathways involved cancer metastasis is EMT, which facilitates cancer cell survival, invasion, and metastasis. This phenomenon is observed in physiological states including embryogenesis and wound healing, and also within several pathologic states [78]. Signaling pathways including STAT3, Notch, and SHH, and growth factors such as bone morphogenic protein (BMP) and transforming growth factor-β (TGF-β), modulate the gene expression of several key EMT transcription factors, including ZEB, Snail and Twist [78]. This results in a phenotypic shift towards a mesenchymal phenotype, with cells undergoing EMT acquiring CSC-like traits [79]. These inlcude features of drug resistance, immunoescape, metastatic capacity, resistance to apoptosis, and anoikis [78]. The EMT enables detachment of cells from the parent tumor, and subsequent invasion into neighboring tissues, thus contributing to the first step of the metastatic cascade—intravasation [78]. Interestingly, induction of EMT with various factors increases cell stemness in immortalized epithelial cells, generating metastasis and cancer cells. Induction of EMT induces a stemness profile: CD44+/HIGH/CD24-/LOW [80], and expression of EMT and CSC markers is correlated with cancer cell invasiveness and metastasis [81]. The observation that induction of EMT confers stem cell traits underscores the connection between CSCs, EMT, and metastasis [82].

### 3.3. JAK/STAT3 Signaling Pathway

The JAK/STAT3 pathway is a critical signaling pathway implicated in many cancer types. There are four JAK family non-receptor tyrosine kinases (JAK1, JAK2, JAK3, and TYK2), and seven members of the STAT family (STAT 1–4, 5A, 5B, and 6). Activation of this pathway results in increased tumorigenicity, metastatic capacity, and enhanced chemotherapy resistance, via EMT. It has been demonstrated that activation of the IL-6/JAK2/STAT3 pathway increases metastasis by facilitating EMT through up-regulation of several transcription factors that induce EMT [81]. Yao et al. demonstrate that NANOG^+^ colorectal cancer cells possess characteristics of both CSCs and EMT. They also observe that the induced-pluripotent stem cell marker NANOG is up-regulated by IGF signaling via STAT3 phosphorylation which regulates both EMT and stemness in these cells [83]. JAK2/STAT3 activation is linked to cancer cell stemness. JAK2/STAT3 activation by oncostatin M, a member of the IL-6 family, facilitates EMT and CSC generation, by increasing Snail and HAS2 levels which act as CD44 ligands causing nuclear accumulation of p-SMAD3 by the STAT3/SMAD3 complex [84,85]. This highlights the JAK/STAT3 signaling cascade as an important therapeutic consideration in developing a multi-target cancer therapy.

## 4. The Tumor Microenvironment and Its Role in Tumor Recurrence and Metastasis

Tumor initiation, progression and metastasis are influenced by genetic and epigenetic factors [86], CSCs, and the dynamic crosstalk between CSCs, cancer cells and other TME components [87] (Figure 3). The TME is a dynamic network comprising cellular and non-cellular components, which orchestrate tumor growth, metastasis, and treatment resistance. The cellular components consist of cancer cells, and stromal cells which include endothelial cells, cancer-associated fibroblasts, and immune cells, primarily consisting of macrophages, microglia, and lymphocytes. Non-cancer cells within the TME contribute to all stages of cancer development and metastasis [88], and include components of the ECM such as laminin, hyaluronan, fibronectin and collagen [87]. Under normal tissue homeostatic conditions, stem cells tightly interact with their surrounding tissue microenvironment to maintain a balance between cell growth and apoptosis. In cancer, these signaling cascades become aberrant and dysregulated, ultimately fostering a tumorigenic *milieu* [89].

### 4.1. Tumor-Associated Macrophages

The immune system within tumors is comprised of multiple components, with tumor-associated macrophages (TAMs) being an integral element. Tumors are infiltrated by cells involved in the innate and adaptive immune systems, reminiscent of the inflammatory environment in non-cancerous tissues [91]. TAMs are abundant within the TME, and are recruited and activated by various chemotactic factors and cytokines. TAMs play a role in facilitating immune escape, cancer progression and metastasis [92]. There are two mechanisms by which these TAMs may be activated. M1, classically activated macrophages, are pro-inflammatory, and facilitate typical inflammatory responses towards cancer cells and pathogens. M2, alternatively activated macrophages, secrete growth factors and anti-inflammatory cytokines, which drive tumor progression and favor tissue repair [93]. TAMs produce factors that stimulate migration which facilitates cancer cell motility and metastasis [94]. TAMs overall help to foster an immunosuppressive TME by producing growth factors and cytokines, that collectively cause the release of inhibitory immune checkpoint proteins by T cells which facilitates certain steps of metastasis [92].

Depending on their surrounding TME, TAMs may either be long-lived embryonically derived tissue-resident macrophages or short-lived monocyte-derived macrophages. The latter are recruited into tumor tissue under the influence of chemokines and growth factors such as macrophage colony-stimulating factor (M-CSF), CCL5, and CCL2 [87,95,96]. In peri-necrotic areas, these factors may stimulate angiogenesis [97].

TAMs may be implicated in cancer metastasis through their role in facilitating the EMT—a process by which epithelial cells morphologically shift towards a mesenchymal phenotype. In the mesenchymal phenotype, these cells lose their apical-basal polarity due to suppression of E-cadherin, and cell–cell junctions which permits motility, thus facilitating invasion and metastasis [98]. The role of TAMs in EMT is further supported by evidence showing that TAMs induce EMT in colorectal cancer to enhance migration, invasion and metastasis through regulation of the JAK2/STAT3/miR-506-3p/FoxQ1 axis [99].

There is interaction between TAMs and CSCs. Macrophages are crucial for the maintenance and retention of hematopoietic stem cells [100,101], and TAMs may play a similar role in maintaining stemness of CSCs [97]. TAMs directly induce stem cell-like characteristics such as chemoresistance in pancreatic ductal adenocarcinoma, by activation of STAT3 signaling [97]. The role of TAMs in influencing CSCs is supported by evidence demonstrating that inhibition of TAMs via CCL2 receptors or M-CSF receptors, decreases the quantity of CSCs and chemoresistance [102]. Under in vitro conditions CD14^+^ macrophages facilitate expansion of CD44^+^ stem cell-like HCC cells, with sphere forming ability used as a marker of stemness, which enhances expression of stemness genes and tumorigenic potential in immunodeficient mice [103]. In this study, TAMs produce IL-6 which drives expansion of CSCs and promotes tumorigenesis. Inhibition of IL-6 signaling with tocilizumab inhibits TAM-driven activity of CD44^+^ cell populations [103]. Given their ubiquity, we speculate that TAMs influence the stemness of CSCs in a range of solid cancers, and thus may contribute to treatment resistance.

### 4.2. The Paracrine Renin-Angiotensin System

Traditionally recognized an endocrine system critical for cardiovascular homeostasis, a self-sufficient paracrine RAS has been identified in cancer [104]. The basic model of the RAS involves the cleavage of angiotensinogen to angiotensin I (ATI), by renin. ATI is then cleaved by angiotensin-converting enzyme (ACE) into its active peptide angiotensin II (ATII), which mediates its actions through interaction with angiotensin receptor 1 (AT_1_R) and angiotensin receptor 2 (AT_2_R) [105,106]. ATII mediates its pathological effects through the interaction with AT_1_R including promotion of cellular proliferation, hypertrophy, inflammation, fibrosis, and oxidative stress [107]. Conversely, interaction of ATII with AT_2_R mediates antagonistic actions including inhibition of cellular proliferation and growth and vasodilation [108]. The contemporary view of the RAS recognizes the importance of several other contributing peptides, including angiotensin III (ATIII) generated from ATII by aminopeptidase A, angiotensin IV (ATIV) generated from ATIII by aminopeptidase M, and angiotensin 1–7 (Ang1–7) generated from ATII by angiotensin-converting enzyme 2 (ACE2). Similar to ATII, ATIII mediates its actions via AT_1_R and AT_2_R, while ATIV mediates its actions via insulin regulating peptidase receptors and results in activation of nuclear factor kappa B (NF-kβ) and vasodilation [108]. Ang1–7, exerts its effects of vasodilation, reduced cellular migration and invasion, anti-fibrotic, anti-thrombotic, anti-hypertrophic, anti-angiogenic effects via the Mas receptor [108,109,110]. Components of the RAS have been demonstrated in numerous cancer types and are localized to the phenotypic CSC populations. These malignancies include glioblastoma [111], head and neck SCC [112,113], oral cavity SCC [113], metastatic melanoma [41,114], renal clear cell cancer [115], and liver metastases from colorectal cancer [116].

The RAS is also expressed by other cells within the TME including immune and stromal cells such as monocytes, macrophages, neutrophils, dendritic cells and T cells, endothelial cells, and fibroblasts. Given the widespread expression of its components, it is not surprising that the paracrine RAS plays an important role in the dynamic intercellular communication between CSCs and their surrounding TME. The RAS promotes cellular proliferation through AT_1_R mediated activation of APK/STAT3, PI3K and AKT signaling pathways [117]. Interaction of ATII with AT_1_R promotes pathological processes including cellular proliferation, migration, invasion, metastasis and inhibition of apoptosis [118]. This ATII/AT_1_R axis contributes to the dissemination of CSCs through its pro-angiogenic actions, resulting in the generation of abnormal, hyperpermeable vessels through up-regulated VEGF-mediated angiogenesis [118]. The ATII/AT_1_R axis may further contribute to tumorigenesis and metastasis, through its role in maintaining an overall hypoxic, acidic, immunosuppressive TME. The pro-fibrotic action of this interaction results in a dense desmoplastic stroma which physically impairs migration of T-cells, and impedes normal vascular perfusion within the tumor. This in conjunction with vasoconstriction leads to the generation of a hypoxic, acidic, immunosuppressive *milieu* [118]. Other mechanisms by which the RAS contributes to tumor invasion and metastasis include its role in ECM remodeling. This occurs through the RAS-mediated up-regulation of MMP-2 and MMP-9 and its pro-inflammatory actions, which up-regulates the expression of cellular adhesion molecules including ICAM-1, VCAM-1 and integrins on endothelial cells; smooth muscle cells; and platelets. These changes facilitate adhesion of tumor cells and their subsequent transmigration [119].

Added complexities to the RAS include the presence of cathepsins B, D and G, proteases which serve as alternatives routes of ATII synthesis, essentially serving as “bypass loops” of the RAS. These alternative signaling pathways enable the RAS to evade single-point inhibitions by traditional RAS inhibitors (RASIs) [120]. Expression of cathepsins B, D and G has been demonstrated in numerous cancer types including glioblastoma [121], head and neck SCC [122,123], malignant melanoma [124], oral tongue SCC [125], and liver metastasis from colorectal cancer [126]. Similar to the expression of components of the RAS within these cancer types, cathepsins B and D are expressed on the phenotypic CSC population, while cathepsin G is expressed by phenotypic mast cells.

### 4.3. Signaling Pathways Converging on the Renin-Angiotensin System

Many signaling pathways converge onto the RAS including the upstream Wnt/β-catenin, and the downstream NADPH oxidase (NOX), reactive oxygen species (ROS), NF-κB, cyclooxygenase 2 (COX2) signaling pathways [120] (Figure 4). The Wnt signaling cascade can be activated by pro-renin receptor (PRR), and is implicated in tumorigenesis, drug resistance, and metastasis. There are several key pathways within the Wnt-signaling cascade—the canonical Wnt pathway, and the non-canonical Wnt pathway [127]. The canonical Wnt pathway, also known as the Wnt/β-catenin pathway, is involved in regulating self-renewal, proliferation, and differentiation of stem cells, and the effects of this pathway are mediated by actions of β-catenin. Under normal conditions, cytoplasmic β-catenin is degraded by a “destruction complex” formed by axin, adenomatous polyposis coli, and glycogen synthase kinase 3β. In the presence of Wnt ligands, this signaling cascade is activated through the binding of Wnt ligands to the Frizzled/low density lipoprotein receptor complex. This leads to the recruitment of dishevelled, an intracellular protein, which inhibits axin mediated phosphorylation of β-catenin, leading to cytoplasmic sequestration of β-catenin. This accumulated β-catenin can then translocate to the nucleus, and form a complex with T-cell transcription factor and lymphoid enhancer-binding factor transcription factor, and this complex regulates transcription of target genes [127,128]. Wnt ligands activating CSCs may be derived from cellular and non-cellular components of the surrounding TME [129]. In CSCs, Wnt signaling influences CSC proliferation through the upregulation of genes including MYC, CCDN1, FOXM1 and YAP/TAZ, and can also up-regulate the expression of various growth factors which contributes to regulation of the surrounding TME [129]. The non-canonical Wnt pathway is implicated in the maintenance of stem cells, inhibition of the canonical Wnt signaling cascade, and is implicated in promoting treatment resistance in CSCs through the activation of P13K-AKT signaling [129]. In this pathway, Wnt ligands interact with the Fx receptor or ROR1k/ROR2/RYK receptors, leading to the activation of either PCP, RTK or intracellular calcium signaling cascades [129]. Both canonical and non-canonical Wnt signaling cascades are involved in invasion and metastasis [129]. Aberrations in the Wnt signaling is well established in colorectal cancer [130], and may be implicated in other malignancies including non-small cell lung [131], breast [132] and hepatocellular [133] cancers. Inhibition of the Wnt signaling cascade thus poses as an elusive mechanism to target CSCs and its interaction with the surrounding TME.

### 4.4. Notch Signaling Pathway

The Notch signaling pathway is a key signaling pathway that is dysregulated in cancer and is implicated in cell proliferation and differentiation, and regulation of apoptosis, all of which contribute to CSC drug resistance [135,136]. The Notch family is comprised of four transmembrane receptors, known as Notch1–4. Notch signaling may have dual actions, acting as both an oncogene and also tumor suppressor gene in skin cancer [137]. However, in many cancers including colon [138], breast [139], pancreatic [140,141], and prostate [142] cancers, Notch appears to function as an oncogene. The Notch signaling pathways interact with other oncogenic pathways including the SHH, Wnt, AKT/mTOR pathways [135]. Up-regulated expression of Notch has been suggested as a potential prognostic marker, with up-regulated expression and poorer outcomes being observed in numerous malignancies including breast [143,144], endometrial [145], cervical [146], gastric [147], pancreatic [148], lung [69], ovarian [72] cancers and glioma [70,71]. Notch signaling has been implicated in CSC-induced treatment resistance with observation of up-regulated expression of drug-efflux pumps in cells also expressing Notch [62]. Interestingly Notch signaling has been proposed as a possible mechanism of chemotherapy resistance. In head and neck SCC, up-regulated expression of Notch is associated with decreased sensitivity to cisplatin, and inhibition of Notch signaling is associated with increased sensitivity to cisplatin treatment [149]. In non-small cell lung cancer, cisplatin treatment increases expression of Notch, and subsequently increases resistance of these cells to doxorubicin and paclitaxel, with inhibition of Notch increasing sensitivity of these cells to doxorubicin and paclitaxel treatment [69]. Notch signaling may also contribute to immunotherapy resistance, with up-regulated expression of notch observed in a trastuzumab-resistant cell line, with Notch inhibition using siRNA associated with increased sensitivity of these cells to transtuzumab [150]. There is compelling evidence supporting the oncogenic role of Notch and its role in contributing to drug resistance, highlighting Notch signaling as an elusive therapeutic target in addressing CSC-induced treatment resistance.

### 4.5. Sonic Hedgehog Signaling Pathway

Hedgehog signaling is initiated by one of three ligands—Sonic, Indian, or Desert hedgehog ligands. Sonic is the most abundantly expressed and potent ligand [151,152]. SHH interacts with transmembrane receptor Patched-1 (PTCH1), which results in loss of inhibition of PTCH1 on Smoothened (SMO) protein. SMO activity decreases interaction of suppressor of fused homology (SUFU) and the GLI family of transcription factors. GLI, the terminal effector protein of the SHH pathway, can then translocate to the nucleus and regulate gene expression [151,152]. There are three GLI isoforms (GLI1-3), with GLI1 inducing gene expression, GL3 supressing gene expression, and GLI2 having bidirectional properties [152]. SHH is involved in normal embryonic development. However, dysregulated expression of Sonic is implicated in pathological states including cancer [151]. In cancer, Sonic is involved in angiogenesis, through up-regulated expression of VEGF and angiopoietins I and II [151], involved in mediating EMT [151], regulating cellular proliferation, migration, invasion [152], and involved in maintaining stemness of CSCs and promoting chemotherapy resistance [153].

Activation of the SHH pathway has been observed in basal cell carcinoma (BCC) [154,155], with expression of GLI1 in basal cells resulting in tumorigenesis [154], and overexpression of PTCH1 [156,157]. The SHH pathway also contributes to the development and progression of pancreatic cancer, through maintaining stemness features of the pancreatic CSCs [158] and enhancing cellular proliferation through NF-κB-mediated activation of the SHH pathway [159]. In pancreatic cancer, the SHH pathway may also contribute to promotion perineural invasion [160] and mediating metastasis with inhibition of the SHH pathway leading to reduced metastasis and lymphangiogenesis [161]. The SHH is also implicated in regulating the surrounding TME and promotes the formation of desmoplastic stroma [162]. Expression of Sonic is up-regulated in the surrounding pancreatic fibroblasts which also demonstrates increased VEGF expression [161], thus alluding to the role of SHH in angiogenesis, and highlighting the intimate role of SHH pathway in the regulation of cancer cells and the surrounding TME. In an in vitro model of prostate cancer, the SHH pathway is involved in cellular proliferation, with inhibition of this pathway with cyclopamine or anti-Sonic antibodies resulting in inhibition of cellular proliferation [163]. In gastric cancer, the SHH pathway is up-regulated and important in the maintenance of stemness of the CSCs and chemoresistance properties of these cells, with inhibition of Sonic by cyclopamine or 5E1 resulting in reduced self-renewal capacity and enhanced response of tumor cells to chemotherapeutic agent oxaliplatin [153].

Modulation of the SHH pathway is thus another important therapeutic consideration in targeting CSCs and the surrounding TME.

## 5. Therapeutic Interventions for Cancer Metastasis and Treatment Resistance

### 5.1. Single-Point Inhibition of Cancer Stem Cells and Signaling Pathways

Given their fundamental role in sustaining tumor growth, treatment resistance, loco-regional recurrence, and distant metastasis, CSCs are potential novel therapeutic target in the treatment of cancer. Proposed strategies for targeting CSCs include inhibition of various signaling pathways expressed and other features of CSCs, or alternatively through modulation of the surrounding TME that sustains CSCs [164].

#### 5.1.1. Targeting the Notch Signaling Pathway

The role of Notch inhibition has been explored in pre-clinical studies and early clinical trials using small molecule inhibitors, known as γ-secretase inhibitors (GSIs), and monoclonal inhibitors. γ-secretase is involved in the Notch signaling cascade and can be targeted using GSIs. In xenograft breast cancer models, Notch signaling inhibition using GSI MK-0752, results in reduction of the breast CSC population, and an enhanced response to the chemotherapeutic agent docetaxel, compared to treatment with docetaxel alone [165]. In a small early clinical trial using GSI MK-0752 for the treatment of patients with breast cancer, reduced CSCs was observed in four of six patients, and two patients had stable disease compared to baseline tumor biopsies. A reduction in ALDH^+^ cells was also observed amongst the participants. However, multiple cycles of treatment were required to yield benefits from this treatment [165]. In a recent pre-clinical study of GSI inhibition of adenoid cystic carcinoma xenograft models treatment with AL101 was associated with inhibition of tumor growth, in models with Notch gain of function mutations [166]. The role of GSIs has been investigated in early clinical trials in patients with leukemia [167], and solid cancers [168] including breast, colorectal, and brain cancers, and have shown no significant effect from single agent use of GSI MK-0752 alone.

Alternatively, the Notch receptors may be targeted by using monoclonal antibodies [136,165]. In a phase Ib clinical trial of metastatic non-squamous small-cell lung cancer, using demcizumab, a monoclonal antibody against delta-like ligand 4-Notch, treatment was associated with a response in 50% of patients, determined by comparison of pre- and post-treatment imaging. One (3%) patient has a complete response, 48% of the patients had a partial response, and 38% had stable disease [169]. However, side effects including pulmonary hypertension and congestive heart failure were observed with its prolonged use [169]. There is scope for further clinical trials exploring the role of GSIs and monoclonal antibodies as a method of targeting CSCs through inhibition of the Notch signaling cascade.

#### 5.1.2. Targeting the Wnt/β-catenin Signaling Pathway

Wnt signaling, which converges on the RAS, has been linked to cancer, treatment resistance, and metastasis. Inhibition of the Wnt signaling cascade is thus a potential therapeutic target for modulation of CSCs. Cyclooxygenase inhibitors, including non-steroidal anti-inflammatory drugs and aspirin, have been shown to have protective effects in cancer, likely through inhibition of the Wnt/β-catenin signaling cascade. In colorectal cancer cell lines, treatment with a COX2 inhibitor increases apoptosis through COX2 independent mechanisms [170], with a further study showing that COX2 selective inhibitor, celecoxib, may exert its COX2 independent anti-neoplastic effects through inhibition of Wnt/β-catenin pathway [171]. The Wnt signaling cascade may be inhibited by targeted therapies including small-molecular inhibitors, or biologics. At present, most studies investigating the use of Wnt inhibitors are pre-clinical studies. In an in vitro and in vivo study of head and neck SCC, inhibition of Wnt ligands inhibits Wnt signaling in vitro, and induces tumor regression in a xenograft model [172], with similar findings observed in a xenograft model of breast cancer [173]. In colorectal cancer, medulloblastoma, neuroblastoma and glioma cells, Wnt inhibition including with COX2 inhibitor celecoxib, enhances the sensitivity of cells and cytotoxic effects of temozolomide, through the down-regulation of O6-methylguanin-DNA methyltransferase, a DNA repair enzyme [174].

Clinical trials assessing the safety and efficacy of Wnt inhibitors are lacking. In a phase I clinical trial using PRI-724, a Wnt signaling inhibitor which inhibits CREB binding protein and β-catenin interaction, PRI-724 had an acceptable safety profile, and was associated with down-regulation of survivin in circulating tumor cells [175]. In a phase Ib study of pancreatic cancer, treatment with PRI-724, 40% of patients had stable disease, 62.5% of patients had a >30% decline in CA19-9 level, however, there was no correlation between serum markers including surviving and clinical outcomes [176].

#### 5.1.3. Targeting the Sonic Hedgehog Signaling Pathway

The SHH pathway, which is implicated in maintenance of CSCs, metastasis, chemotherapy resistance and regulating the TME, can be modulated by vismodegib (GDC-0449), a cyclopamine-derived competitive SMO agonist [152]. At present, this medication is FDA approved only for the treatment of BCC. In a multicentre non-randomized study, of individuals with locally advanced or metastatic BCC, treatment with vismodegib yielded partial responses in all patients with metastatic responses, with 73% of patients exhibiting tumor shrinkage [177]. In those with locally advanced disease, treatment with vismodegib yielded complete response (absent of residual BCC) in 21% of patients, while the majority of patients exhibited tumor shrinkage [177]. Similar results were demonstrated in a single-arm multicentre study of the use of vismodegib for the treatment of locally advanced and metastatic BCC. In this study, vismodegib had a good safety profile, and a complete response was seen in 33.4% of those with locally advanced and 4.8% with metastatic disease, and partial response observed in 68.5% and 36.9% of those with locally advanced and metastatic BCC, respectively [178]. Vismodegib, albeit not yet FDA approved, has been explored as a potential inhibitor in other malignancies. Interestingly, in a phase Ib/II study of vismodegib in conjunction with the chemotherapeutic agent gemcitabine, no significant difference between progression-free survival or overall survival was observed compared to placebo [179]. Similarly, no significant difference in overall survival or progression-free survival was observed in a phase II clinical trial of vismodegib treatment, in addition to conventional treatment with bevacizumab, for metastatic colorectal cancer [180]. These varied outcomes could be associated with sample size, or intrinsic study features such as dosing and duration of treatment, or could also be explained by the numerous regulatory pathways involved in regulating CSCs and the surrounding TME.

Sonidegib (LDE-225) is another inhibitor of the SHH pathway, through its action as a SMO agonist [152]. It is FDA approved for the treatment of BCC. A phase II trial shows complete or partial response in 43% of patients with locally advanced BCC, and 15% of those with metastatic BCC taking 200 mg of sonidegib, and 38% of those with locally advanced BCC, and 17% of those with metastatic BCC taking 800 mg sonidegib once daily, with less severe side effects in the 200 mg group [181]. Early clinical trials have exploring the use of sonidegib in other malignancies including medulloblastoma [182], small cell lung [183], triple negative breast [184] and prostate [185] cancers, have shown that sonidegib is largely well-tolerated. Interestingly in medulloblastoma, those who responded to sonidegib treatment expressed HH genes, while those who did not express HH genes did not respond [182]. However, further phase II trials are required to explore the efficacy of these treatments.

5E1, a monoclonal antibody against the Sonic ligand has been used in pre-clinical models of cancer with promising results [152]. In a xenograft model of cervical cancer, 5E1 treatment enhances the effect of treatment with adjuvant chemotherapy and radiotherapy, but has no effect as a single agent [186]. Other xenograft models of cancer have shown that 5E1 treatment enhances the effect of chemotherapy in gastric cancer [153], suppresses tumor growth and increases survival in medulloblastoma [187], and reduces tumor growth and metastasis in pancreatic cancer [161]. Further clinical studies are warranted to determine the safety and efficacy of this monoclonal antibody in the treatment of cancer.

#### 5.1.4. Strategies to Overcome Targeting Multi-Drug Resistance

In MDR, CSCs may simultaneously exhibit resistance to multiple structurally different drugs and addressing mechanisms contributing to MDR may enhance chemo-sensitivity of CSCs. One key mechanism driving MDR in CSCs is the presence of ABC-transporters or “drug efflux pumps”. Inhibition of HSP90, which regulates expression of the Pgp family of ABC-transporters, is associated with downregulated expression of Pgp, and enhanced chemo-sensitivity in colorectal cells in vitro [188]. The use of nanomedicines is an emerging method to target MDR. Nanomedicine is a novel strategy to deliver treatment directly to cancer cells by way of polymeric nanoparticles, liposomal particles, inorganic nanoparticles, or hybrid nanoparticles [75]. The emerging concept of targeted therapy using nanomedicine has several elusive features including facilitating the precise treatment of cancer. Through the enhanced permeability and retention effect, or active transport via endothelial cell pathways, nanomedicine allows precise treatment of cancers, by locally increasing the drug concentration within tumor tissue, thus increasing treatment efficacy and decreasing MDR. This may also decrease the off-target effects of drugs, and effects of these drugs on non-tumor cells. Specificity of nanomedicines for cancer cells may be achieved through the use of specific ligands, aptamers, peptides, and antibodies specific for cancer surface biomarkers [75]. Nanomedicines may also provide a modality to co-deliver multiple combinations of drugs [75].

### 5.2. Single-Point Inhibition of the Tumor Microenvironment

The TME is critical in regulating CSC growth, treatment resistance and metastasis. The TME may be modulated through inhibiting angiogenesis, inhibiting macrophage recruitment and activation, targeting tumor cell-derived exosomes, enhancing the anti-tumor axis of the local immune system, and by targeting tissue hypoxia and cancer-associated fibroblasts [189].

Immune activity is variable within the TME of different tumors. Some show minimal inflammation, and others exhibit extensive infiltration of immune cells of both the innate and adaptive arms. Several methods of targeting the tumor immune system have been proposed. These include inhibition of macrophage recruitment into the tumor, targeting factors that drive chronic inflammation and the pro-tumor factors released by local lymphocytes, enhancement of anti-tumor immune mechanisms, and inhibition of macrophage differentiation into TAMs [189]. The immune system may be modulated by immunotherapeutic agents, which has been shown to improve survival outcomes in cancers including malignant melanoma, non-small cell lung, and bladder cancers [190]. Anti-programmed cell death 1 antibodies, which may be used alone or in addition to other immunotherapies, have been explored as a potential immunotherapeutic agent [191]. Other immunotherapy agents look at exploiting the ability of CSCs to evade apoptosis, programmed cell death, and targeting ferroptosis, necroptosis, and autophagy [192].

Tumorigenesis is reliant on the formation of new aberrant blood vessels for nutrient delivery and dissemination of tumor cells, thus one strategy to target cancer and the TME is through inhibiting angiogenesis. Bevacizumab, an inhibitor of VEGF-A, is commonly used with 5-fluorouracil, and has been used in the treatment of several cancers including gastric, colorectal and cervical adenocarcinoma [193]. In a randomized controlled trial (RCT) on the use bevacizumab, in addition to 5-fluorouracil-based chemotherapy, for metastatic colorectal cancer, treatment with bevacizumab was associated with increased survival of 20.3 months, compared to 15.6 months in the placebo group [194]. However, the use of anti-angiogenic therapies may create areas of hypoxia within the TME and lead to a propensity for a CSC phenotype [193]. The γ-secretase inhibitor RO4929097 reduces CSCs in glioma, but also increases treatment resistance [195]. Remodeling of the ECM is required for angiogenesis, by the action of MMPs. However, trials investigating MMP inhibitors have not shown a significant impact on overall survival [196]. In an RCT investigating treatment of small cell lung cancer, the MMP inhibitor marimastat does not improve survival, and negatively impacts quality of life, with no significant difference in time to progression compared to the placebo group [197]. It is important to consider the vital role of MMPs in normal cellular functions, and their inhibition may negatively impact normal cellular processes [198].

Glioblastoma stem cells (GCSs) operate within a tissue microenvironment functionally similar to the *niche* of hematopoietic stem cells (HSCs) within normal bone marrow [199]. Transcription factors HIF-1α and HIF-2α are expressed in both microenvironments, highlighting that hypoxia is critical for the maintenance of the HSC and GCSs [199]. Within both GCS and HSC tumor microenvironments, chemokine chemoattractant stromal-derived factor-1α (SDF-1α) is expressed. SDF-1α binds to C-X-C receptor type 4 (CXCR4) which is expressed on GCSs and HSCs [199]. In hypoxic conditions, SDF-1α and CXCR4 expression in both bone marrow and glioblastoma is increased via HIF-1α [200]. Binding of SDF-1α to CXCR4 to critical for cell survival in glioblastoma, by maintaining GCSs in a slowly dividing state, which thus reduces sensitivity to treatments including radiotherapy and chemotherapy which target rapidly dividing cells. This highlights the SDF-1α-CXCR4 receptor-ligand interaction as being a possible therapeutic target for GSCs. CXCR4 may be inhibited by plerixafor (AMD3100), and this may shift CSCs out of their slow-division state within glioblastoma and AML, thus enhancing their susceptibility to other treatment modalities [201,202]. Targeting CXCR4 has also been proposed as a potentially treatment strategy in patients who have minimal residual disease who undergo HSC transplantation.

### 5.3. Targeting the Renin-Angiotensin System

The RAS, which is routinely targeted by various RAS inhibitors (RASIs) in the treatment of hypertension and other cardiovascular diseases, is another potential therapeutic target (Figure 5). Drugs inhibiting the RAS include β-blockers, which inhibit conversion of pro-renin into active renin, ACE inhibitors (ACEIs), and angiotensin receptors blockers (ARBs). Various epidemiological studies have highlighted a largely protective role of RASIs in cancer with reduced cancer risk and/or improved cancer outcomes [203,204,205,206,207,208,209,210,211,212,213,214,215,216,217,218,219,220,221,222,223,224,225,226,227,228,229,230,231,232,233,234,235,236,237], although a small cohort of epidemiological studies show no effect [238,239,240,241,242,243,244,245,246,247,248,249,250], or a harmful effect [251,252]. This can be attributed to variations in study population, sample size, duration of follow-up, but could also be explained by the presence of inbuilt redundancies within the RAS, thus reducing the effect of RASIs. These include the up-stream Wnt/β-catenin, and the down-stream NOX-ROS-NF-κB-COX2 convergent signaling pathways, and the presence of protease enzymes cathepsins B, D and G which serve as bypass loops of the RAS, allowing alternative methods of ATII biosynthesis (Figure 5). Effective inhibition of the RAS may thus require inhibition of this system at multiple points simultaneously using multiple drugs.

### 5.4. Photosensitizer-Based Therapies

Photosensitizer-based therapies are methods of delivering localized light radiation to cancer cells, thus creating localized tissue damage. Photosensitizer therapies include photodynamic therapy (PDT) and photothermal therapy (PTT) [253]. Photosensitizing agents can be delivered through intravenous or topical administration, and following administration of this photosensitizer, light of a certain wavelength may be administered to affect the target site. PTT employs the use of photothermal agents, which when irradiated by light, absorb energy from photons, which creates local thermal-induced tissue damage [253]. Conversely, in PDT, when the photodynamic agent is irradiated, absorption of photons, leads to the generation of ROS which subsequently creates localized cytotoxic cellular damage [253]. Use of photosensitizer therapies for the treatment of cancer has been gaining traction over the past several decades, although dermatological contraindications limit their broad use [253]. Photosensitizer therapies may be used in conjunction with other current treatment modalities including chemotherapy and immunotherapy. When used in conjunction with chemotherapy, PDT may enhance chemo-sensitivity through the inhibition of Pgp drug efflux pumps by the generation of ROS [253]. Next-generation photosensitizing molecules used with targeted nanoparticles capable of delivering multimodal drug therapies directly to the TME may increase treatment efficacy, reduce MDR, and reduce non-target tissue adverse effects. However, further clinical studies are warranted to explore the efficacy and safety of this.

### 5.5. A Multi-Target Approach in the Treatment of Cancer

CSC growth and communication with the surrounding TME is regulated by numerous pathways which together orchestrate treatment resistance and metastasis. Thus, a single point inhibition without simultaneous modulation of other involved pathways may explain the partial, and at times negative, response shown in these pre-clinical and early clinical studies. An effective approach to the treatment of cancer may thus require a multi-modal approach involving multi-point inhibition of CSCs; components of the TME and immune system; the RAS, its bypass loops consisting of enzymes such as cathepsins B, D and G, and its convergent signaling pathways. These include the upstream Wnt/β-catenin and downstream NOX-ROS-NF-κB-COX2 cascades, and other signaling pathways such as the Notch and SHH pathways (Figure 5).

We have recently presented results of a phase I clinical trial on recurrent glioblastoma by targeting CSCs in by influencing the TME to modulate the RAS using multiple medications (propranolol, aliskiren, cilazapril, celecoxib, curcumin with piperine, aspirin, and metformin) [254]. The treatment was well tolerated with low side-effects, and the quality of life and performance status of the participants were preserved during treatment with an overall median survival 19.9 months [254]. The increased survival of 5.3 months, although not statistically significant, was encouraging, and warrants further investigation.

The CUSP9 protocol proposed in 2013 [255] was modified to CUSP*, uses aprepitant, artesunate, auranofin, captopril, celecoxib, disulfiram, itraconazole, sertraline and ritonavir. Each drug inhibits the growth-enhancing pathways of 17 different systems within the TME of glioblastoma [256]. In 2019 this strategy was employed on patient-derived glioblastoma stem cells (GSC) from 15 patients. CUSP9 with temozolomide exerted a combination effect compared to individual drug effects. Half of the GSCs demonstrated high sensitivity to this combination. Interestingly, CUSP9 significantly reduces Wnt-activity, a critical stem cell pathway employed by GSCs [257].

The phase Ib/IIa CUSP9v3 trial administered the nine repurposed drugs with low dose temozolomide, to patients with progressive or recurrent glioblastoma. This treatment regimen was overall safe and well tolerated, and there was a progression-free survival of 50% at 12 months [258]. Itraconazole, ritonavir, captopril, and temozolomide are the agents most often required dose modification for side effects, most commonly diarrhoea, nausea, fatigue, headache, and ataxia [258].

This multimodal targeting of CSCs and the TME in the treatment of cancer is an emerging concept, and there is a paucity of clinical trials exploring this strategy with early clinical trials showing encouraging results.

## 6. Conclusions

Cancer is the most common cause of death, being responsible for nearly one third of all deaths, with most cancer-related deaths attributable to metastasis. Cancer poses huge economic and personal burden, with an estimate of over 19 million new cases of cancer and 10 million cancer-related deaths globally in 2020 [259]. The mainstay treatments for cancer—surgery, radiotherapy, chemotherapy, and/or immunotherapy are only partially effective, with treatment resistance and failure, and metastasis driving cancer mortality.

The hierarchical model of cancer underscores the presence of a subset of tumorigenic, self-renewing CSCs sitting at the apex of a cellular hierarchy. This small subpopulation of CSCs is thought to sustain tumor growth, and contribute to treatment resistance, tumor recurrence, and metastasis. CSCs are thus a compelling therapeutic target for the effective treatment of cancer. Various proposed mechanisms underscore treatment resistance and thus metastasis within CSCs. These mechanisms include tumor dormancy, the presence of ABC transporters, expression of lncRNA, and the upregulated expression of DNA repair mechanisms, free radical scavenger systems and anti-apoptotic proteins. Furthermore, the concept of treatment resistance is further complexified by the emerging phenomenon of MDR which highlights a concept whereby CSCs are resistance to multiple non-related drugs.

There is dynamic communication between CSCs and the surrounding TME to facilitate the formation of a tumorigenic *niche* that promotes treatment resistance and metastasis. This pro-tumorigenic TME is regulated through components of the immune system, the paracrine RAS and its bypass loops consisting of enzymes cathepsins B, D and G, and convergent signaling pathways including the upstream Wnt/β-catenin, and down-stream NOX-ROS-NF-κB-COX2 convergent signaling pathways, and other signaling pathways including Notch and SHH pathways.

An effective treatment for cancer requires appreciation of these key mechanisms underscoring treatment resistance and metastasis within CSCs and the TME. Numerous pre-clinical and early phase clinical studies exploring the role single-point inhibition of signaling pathways involved in CSC growth, differentiation, maintenance, and treatment resistance, including the Wnt/β-catenin and SHH pathways, have shown a largely partial response. Similar partial responses are yielded in studies exploring single point inhibition of the TME through various immunotherapeutic agents, inhibitors of angiogenesis, and RASIs. The effectiveness of a single point inhibition of CSCs and the TME may be mitigated by the presence of multiple pathways contribute to cancer growth, treatment resistance and metastasis. An effective treatment of cancer is likely to require simultaneous multi-point inhibition of the key signaling pathways implicated in regulation of CSCs and the TME, which are implicated in treatment resistance and metastasis. However, many of these proposed treatments are still in their pre-clinical and early clinical trial stage, and further studies are warranted to fully assess the safety and efficacy of these treatments

The effective treatment of cancer is likely to require a multimodal treatment approach using pre-existing treatment modalities such as surgery, radiotherapy, chemotherapy and/or immunotherapy, in conjunction with multiple drugs targeting various implicated signaling pathways. Other novel treatment strategies that may be an elusive adjuvant treatment for cancer including the use of nanomedicines to provide targeted delivery of multiple drugs to localized tumor tissue, and photo-sensitizer therapies. There are a limited number of clinical studies exploring this multi-modal, multi-drug treatment approach. A phase I clinical trial on recurrent glioblastoma, shows that the treatment is well-tolerated with minimal side effects, and a non-significant but encouraging increase in survival by 5.3 months. A similar multi-drug approach is used in the CUSP* protocol for the treatment of progressive or recurrent, and encouraging results are also observed, with a progression-free survival of 50% at 12 months. While these studies have yielded promising results, further larger clinical studies are required to explore long term outcomes, and outcomes in other cancers.

Improved understanding of the mechanisms driving treatment resistance and metastasis within the CSC and surrounding TME is critical for the development of an effective cancer treatment. The concept of cancer treatment resistance and metastasis is a complex phenomenon that is driven by a multitude of signaling pathways and mechanisms. It is becoming clear that single point inhibition of these pathways may be made redundant, and may explain the partial responses observed with current treatments and studies, due to the presence of alternative mechanisms which can yield the same outcome. Thus, an effective treatment for cancer may require a broader approach utilizing a multi-modal strategy encapsulating multi-step inhibition of the pathways which regulate CSCs and the TME, *in lieu* of the long-standing pursuit of a ‘silver bullet’ single-target approach to cancer.

## 7. Patents

ST is an inventor of the patents Cancer Diagnosis and Therapy (PCT/NZ2015/050108, AUS/2012302419, JAP/2017528398, and US/0281472), Cancer Therapeutic (PCT/NZ2018/050006), Novel Pharmaceutical Compositions for Cancer Therapy (PCT/NZ2019/050087), Treatment of fibrotic conditions (PCT/NZ2016/050187), Treatment of vascular anomalies (PCT/NZ2017/050032), and Methods and compositions for the treatment of hemangioma (PCT/NZ2021/050012).

## Figures and Tables

**Figure 1 biomedicines-10-02988-f001:**
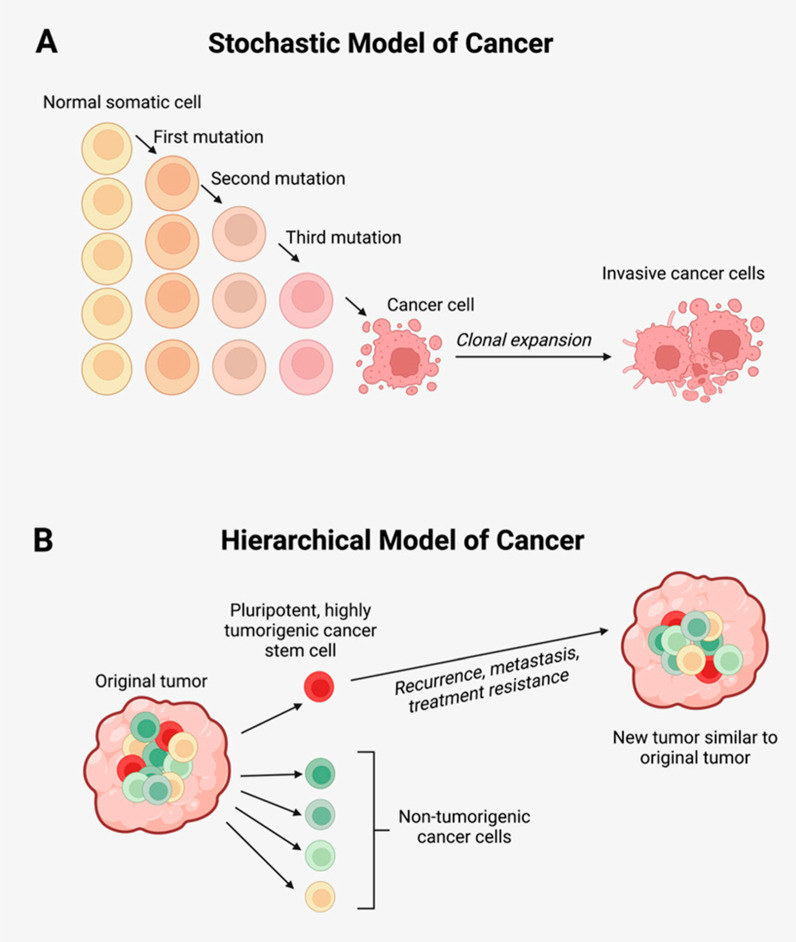
(**A**) The stochastic model of cancer proposes that a normal somatic cell accumulates oncogenic mutations in a stepwise manner and becomes a cancer cell that undergoes clonal expansion to form a tumor. (**B**) The hierarchical model of cancer proposes the presence of a highly tumorigenic cancer stem cell (CSC) sitting atop the tumor cellular hierarchy and divides asymmetrically to form non-tumorigenic cancer cells that form the bulk of the tumor, and identical CSCs that form new tumors like the original tumor. Adapted from the Atlas of Extreme Facial Cancer, Springer Nature [9]. Diagram recreated with BioRender.com, accessed on 1 November 2022.

**Figure 2 biomedicines-10-02988-f002:**
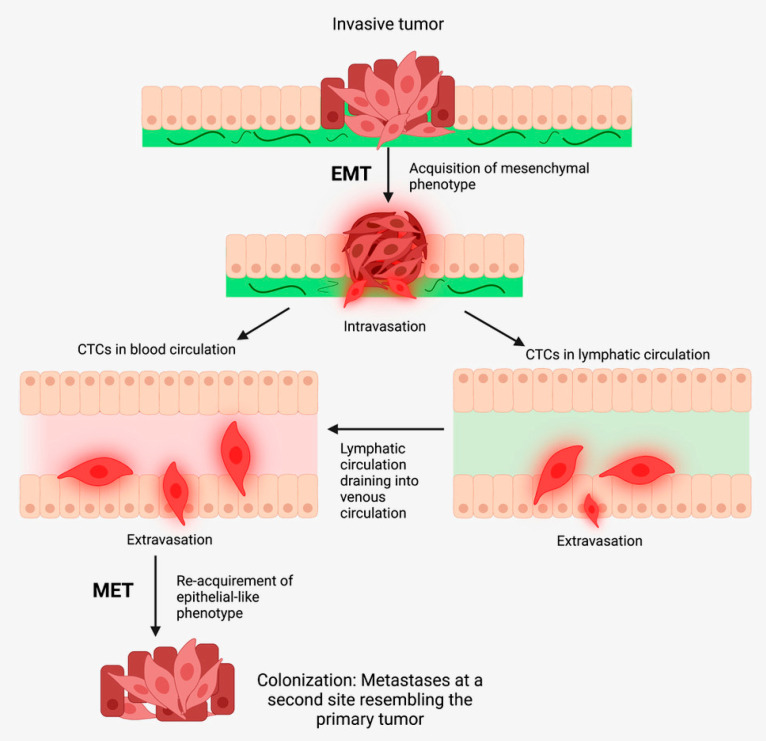
In cancer metastasis, tumor cells undergo epithelial-mesenchymal transition (EMT) to form mesenchymal-like cells, which undergo intravasation to enter the blood and/or lymphatic circulations as circulating tumor cells (CTCs). CTCs then undergo mesenchymal-epithelial transition (MET) and extravasate into distant tissue sites, where metastases may form. Adapted from the Atlas of Extreme Facial Cancer, Springer Nature, 2022 [9]. Diagram recreated with BioRender.com, accessed on 1 November 2022.

**Figure 3 biomedicines-10-02988-f003:**
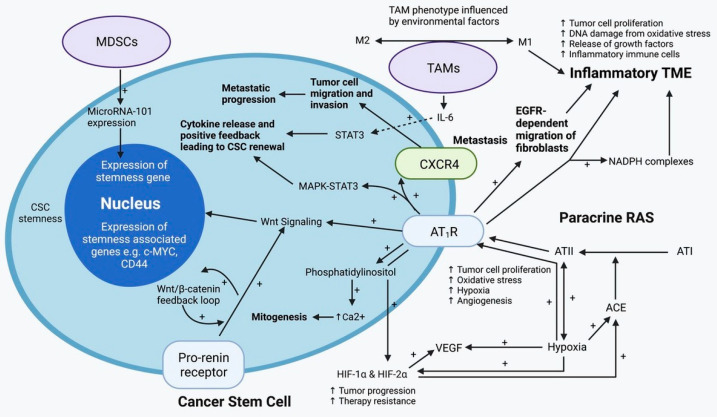
A cancer stem cell (CSC) residing within the tumor microenvironment (TME) which is regulated by the paracrine renin–angiotensin system (RAS) and the immune system. The active end-product of the paracrine RAS—angiotensin II (ATII), activates ATII receptor 1 (AT_1_R) resulting in increased tumor cell proliferation, oxidative stress, hypoxia and angiogenesis, and inflammation—the hallmarks of cancer. This contributes to an inflammatory TME by increasing the number of inflammatory cells, partly by increasing the number of NADPH complexes, leading to tumor cell proliferation, DNA damage from oxidative stress, and release of growth factors. AT_1_R also activates phosphatidylinositol signaling which increases cytosolic Ca2+ to promote mitogenesis. Hypoxia increases paracrine RAS activity by up-regulating angiotensin-converting enzyme (ACE) and hypoxia-inducible factor 1α (HIF-1α) and HIF-2α expression, which increase tumor progression and treatment resistance. HIF-1α, HIF-2α, and hypoxia also increase the expression of vascular endothelial growth factor (VEGF) which promotes angiogenesis. Binding of AT_1_R to C-X-C chemokine receptor type 4 (CXCR4) promotes tumor cell migration and invasion, leading to metastatic spread. AT_1_R, via MAPK-STAT3 signaling, contributes to a cytokine release that leads to CSC renewal. AT_1_R signaling also contributes to the migration of fibroblasts in an epidermal growth factor receptor (EGFR)-dependent fashion. AT_1_R signaling and the pro-renin receptor, acting in a feedback loop with Wnt/β-catenin, increase Wnt signaling which promotes CSC stemness by up-regulating stemness-associated markers. Myeloid-derived suppressor cells (MDSCs) promote CSC characteristics by increasing the expression of microRNA-101 that induces expression of stemness-related genes in CSCs. Under the influence of the TME, polarization of tumor-associated macrophages (TAMs) within the TME, changes from the M1 to M2 phenotype. M2 TAMs induce the proliferation of CSCs via interleukin-6 (IL-6)-induced activation of STAT3, leading to cytokine release and positive feedback that contribute to CSC renewal. Abbreviations: ATI, angiotensin I; AT_2_R, ATII receptor 2; Ang1–7, angiotensin 1–7; ATIII, angiotensin III; MAPK, mitogen-activated protein kinase. Adapted from the Journal of Histochemistry and Cytochemistry [90]. Diagram recreated with BioRender.com, accessed on 1 November 2022.

**Figure 4 biomedicines-10-02988-f004:**
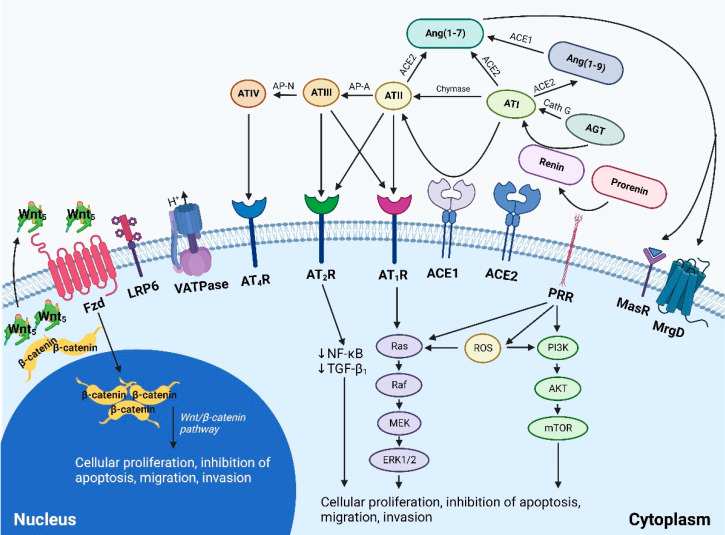
A schema showing the effect of the paracrine renin–angiotensin system (RAS) and its convergent signaling pathways on the tumor microenvironment to influence cellular proliferation, invasiveness, and cell survival in cancer development. The RAS interacts with downstream pathways, such as the Ras/RAF/MEK/ERK (light blue) pathway and the PI3K/AKT/mTOR (dark blue) pathway, and the up-stream Wnt/β-catenin pathway (intermediate blue) that influence cellular proliferation, migration, inhibition of apoptosis, migration, and invasion (see text). PRR, pro-renin receptor; LRP6, low-density lipoprotein receptor-related protein; Fzd, frizzled receptor; Cath G, cathepsin G; Cath B, cathepsin B; Cath D, cathepsin D; ACE1, angiotensin-converting enzyme 1; ACE2, angiotensin-converting enzyme 2; ADP, adenosine diphosphate; AGT, angiotensinogen; ATP, adenosine triphosphate; Ang(1–7), angiotensin (1–7); Ang(1–9), angiotensin (1–9); AP-A, aminopeptidase-A; NEP, neutral endopeptidase; AP-N, aminopeptidase-N; ATI, angiotensin I; ATII, angiotensin II; ATIII, angiotensin III; ATIV, angiotensin IV; AT_1_R, angiotensin II receptor 1; AT_2_R, angiotensin II receptor 2; AT_4_R, angiotensin II receptor 4; MrgD, Mas-related-G protein coupled receptor; MasR, Mas receptor; mTOR, mammalian target of rapamycin; NF-κB, nuclear factor kappa B; TGF-β_1_, transforming growth factor-β_1_; V-ATPase, vacuolar H^+^-adenosine triphosphate. Adapted from Cancers [134]. Diagram recreated with BioRender.com, accessed on 1 November 2022.

**Figure 5 biomedicines-10-02988-f005:**
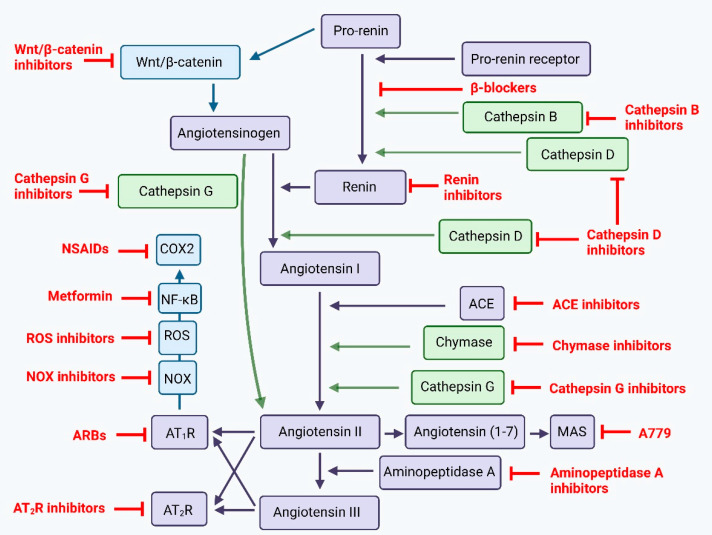
The renin-angiotensin system and its bypass loops and converging signaling pathways can be targeted at different points. The renin-angiotensin system (black) regulates blood pressure, stem cell differentiation, and tumor development. Bypass loops in the system involving cathepsins and chymase (green) provide redundancy, while convergent inflammatory and development signaling pathways (blue) have multiple roles and effects. Multiple points of the pathway can be targeted by specific inhibitors (red). ACE, angiotensin converting enzyme; ARBs, AT_1_R blockers; ROS, reactive oxygen species; NSAIDS, non-steroidal anti-inflammatory drugs. Adapted from Frontiers in Oncology [120]. Diagram recreated with BioRender.com, accessed on 1 November 2022.

## Data Availability

Not applicable.

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
