# Peer review of "Cancer Metastasis and Treatment Resistance: Mechanistic Insights and Therapeutic Targeting of Cancer Stem Cells and the Tumor Microenvironment"

_biomedicines, 2022, doi:10.3390/biomedicines10112988_

Round 1
Reviewer 1 Report
This review manuscript is a fine piece of work. It describes the present state of our understanding of metastasis and therapy resistance of cancers. I think it is an important contribution in the field as a starting point for scientists and students to get insight in the clinically highly relevant topics in cancer progression.
I have only three minor comments:
1. Page 3, lines 12-13: basement membrane is an outdated concept that was created when EM was introduced in the sixties of last century. It was then considered to be a membrane structurs, whereas it is ECM consisting of proteins. The accurate term is 'basal lamina' or 'lamina basdalis'. Please correct.
2. Page 7, lines 14-15: Oncostatin M.....family. This is not a proper sentence. Please correct.
3. In Figure 3, CXCR4 is mentioned but it is not discussed in the text, whereas the CXCR4-SDF-1a axis as well as the CD44-osteopontin axis are important binding mechanisms for CSCs at least in leukemia and glioblastoma, but I am sure in other types of tumors as well. The inhibitor plerixafor of CXCR4 is being tested in clinical trials to remove CSCs from their niches to make them vulnerable for therapy. See references: Hira et al (2020) Biology 9:31; Bernasconi and Borsani (2019) J Oncol 2019:8323592.
These binding axes and the therapeutic interference in this binding with sebsequent more effective radio- and/or chemotherapy needs to be discussed.
Author Response
Authors’ Response to Reviewer 1’s Comments
Comments and Suggestions for Authors
This review manuscript is a fine piece of work. It describes the present state of our understanding of metastasis and therapy resistance of cancers. I think it is an important contribution in the field as a starting point for scientists and students to get insight in the clinically highly relevant topics in cancer progression.
Authors’ response: We are very grateful for your support.
I have only three minor comments:
- Page 3, lines 12-13: basement membrane is an outdated concept that was created when EM was introduced in the sixties of last century. It was then considered to be a membrane structurs, whereas it is ECM consisting of proteins. The accurate term is 'basal lamina' or 'lamina basdalis'. Please correct.
Authors’ response: Thank you for your suggestion which we have included in our revised manuscript.
- Page 7, lines 14-15: Oncostatin M.....family. This is not a proper sentence. Please correct.
Authors’ response: Thank you for highlighting this error which has been corrected in our revised manuscript.
- In Figure 3, CXCR4 is mentioned but it is not discussed in the text, whereas the CXCR4-SDF-1a axis as well as the CD44-osteopontin axis are important binding mechanisms for CSCs at least in leukemia and glioblastoma, but I am sure in other types of tumors as well. The inhibitor plerixafor of CXCR4 is being tested in clinical trials to remove CSCs from their niches to make them vulnerable for therapy. See references: Hira et al (2020) Biology 9:31; Bernasconi and Borsani (2019) J Oncol 2019:8323592.
These binding axes and the therapeutic interference in this binding with sebsequent more effective radio- and/or chemotherapy needs to be discussed.
Authors’ response: Thank you for your insightful comments and for highlighting this interesting area. We have included the useful information on these topics in our revised manuscript.
Reviewer 2 Report
The Review article entitled “Cancer Metastasis and Treatment Resistance: Mechanistic Insights and Therapeutic Strategies” systematically reviewed, Cancer Stem Cells and Metastasis, as well as various pathways involved in Metastasis, increased tumorigenicity, metastatic capacity, and enhanced chemotherapy resistance, etc. The authors have also highlighted how these pathways might be targeted to overcome drug resistance and improve therapeutic intervention. But authors some key emerging therapeutic interventions for cancer. The article has many grammatical and sentence errors, and the language organization needs to be improved. For these reasons, I conclude that the paper should undergo minor revision
1. The introduction is good but very general in nature. Authors need to provide more insight into Treatment resistance, notably chemotherapy resistance and MDR. Explain MDR and current methods to overcome MDR like the targeted release of the drug helps to overcome MDR. More elaborately.
2. Authors need an added section on emerging therapeutic interventions like photodynamic and photothermal, gene and hormone therapy in combination treatment which helps in overcoming MDR and improving therapeutic prognosis.
3. Role of HSP and the immune system in acquiring MDR needs to be discussed and methods to overcome it needs to elaborate.
4. Authors need to improve the references by citing recent references like
https://doi.org/10.1016/j.bbagen.2022.130113
https://doi.org/10.1007/s10555-022-10024-8
https://doi.org/10.1016/j.sjbs.2022.03.005
5. Title seems very general. Please ensure all keywords are used in the abstract and the title.
6. Typographical errors can be avoided. The language and grammar used throughout the manuscript need to be improved.
7. Conclusion need to improve by providing a better perspective among the existing treatment and evolving ones, how there can be improved and used in combination in the future to obtain the best intervention for cancer
Author Response
Authors’ Response to Reviewer 2’s Comments
Comments and Suggestions for Authors
The Review article entitled “Cancer Metastasis and Treatment Resistance: Mechanistic Insights and Therapeutic Strategies” systematically reviewed, Cancer Stem Cells and Metastasis, as well as various pathways involved in Metastasis, increased tumorigenicity, metastatic capacity, and enhanced chemotherapy resistance, etc. The authors have also highlighted how these pathways might be targeted to overcome drug resistance and improve therapeutic intervention. But authors some key emerging therapeutic interventions for cancer. The article has many grammatical and sentence errors, and the language organization needs to be improved. For these reasons, I conclude that the paper should undergo minor revision
- The introduction is good but very general in nature. Authors need to provide more insight into Treatment resistance, notably chemotherapy resistance and MDR. Explain MDR and current methods to overcome MDR like the targeted release of the drug helps to overcome MDR. More elaborately.
Authors’ response: Thank you for your support and for highlighting MDR. We have now included this information in our revised manuscript and trust it is satisfactory.
- Authors need an added section on emerging therapeutic interventions like photodynamic and photothermal, gene and hormone therapy in combination treatment which helps in overcoming MDR and improving therapeutic prognosis.
Authors’ response: Thank you for your suggestions. We have now added these emerging therapeutic interventions to overcome MDR in our revised manuscript. We trust that this is now satisfactory.
- Role of HSP and the immune system in acquiring MDR needs to be discussed and methods to overcome it needs to elaborate.
Authors’ response: Thank you for your suggestion which we have included in our revised manuscript.
- Authors need to improve the references by citing recent references like
https://doi.org/10.1016/j.bbagen.2022.130113
https://doi.org/10.1007/s10555-022-10024-8
https://doi.org/10.1016/j.sjbs.2022.03.005
Authors’ response: Thank you for highlighting these relevant articles which we have now cited in our revised manuscript.
- Title seems very general. Please ensure all keywords are used in the abstract and the title.
Authors’ response: Thank you for your insightful comment. We have now amended the title of our article to better reflect and ensure the key words are reflected in the title and abstract.
- Typographical errors can be avoided. The language and grammar used throughout the manuscript need to be improved.
Authors’ response: Thank you for your advice. We have now carefully proof-read the manuscript and corrected the errors.
- Conclusion need to improve by providing a better perspective among the existing treatment and evolving ones, how there can be improved and used in combination in the future to obtain the best intervention for cancer
Authors’ response: Thank you for your insightful comment. We have now broadened our conclusion to include the important points you have raised. We have discussed using a multi-target strategy as a potential strategy to improve effectiveness of cancer treatment.